# Beyond Activation: Characterizing Microglial Functional Phenotypes

**DOI:** 10.3390/cells10092236

**Published:** 2021-08-28

**Authors:** Julia Lier, Wolfgang J. Streit, Ingo Bechmann

**Affiliations:** 1Institute of Anatomy, University of Leipzig, 04109 Leipzig, Germany; Ingo.Bechmann@medizin.uni-leipzig.de; 2Department of Neurology, University of Leipzig, 04109 Leipzig, Germany; 3Department of Neuroscience, University of Florida College of Medicine, Gainesville, FL 32611, USA; pschorr@ufl.edu

**Keywords:** microglia, IBA1

## Abstract

Classically, the following three morphological states of microglia have been defined: ramified, amoeboid and phagocytic. While ramified cells were long regarded as “resting”, amoeboid and phagocytic microglia were viewed as “activated”. In aged human brains, a fourth, morphologically novel state has been described, i.e., dystrophic microglia, which are thought to be senescent cells. Since microglia are not replenished by blood-borne mononuclear cells under physiological circumstances, they seem to have an “expiration date” limiting their capacity to phagocytose and support neurons. Identifying factors that drive microglial aging may thus be helpful to delay the onset of neurodegenerative diseases, such as Alzheimer’s disease (AD). Recent progress in single-cell deep sequencing methods allowed for more refined differentiation and revealed regional-, age- and sex-dependent differences of the microglial population, and a growing number of studies demonstrate various expression profiles defining microglial subpopulations. Given the heterogeneity of pathologic states in the central nervous system, the need for accurately describing microglial morphology and expression patterns becomes increasingly important. Here, we review commonly used microglial markers and their fluctuations in expression in health and disease, with a focus on IBA1 low/negative microglia, which can be found in individuals with liver disease.

## 1. Introduction

Interest in microglial function has been growing exponentially since the 1990s. Today, there is almost no disease in the CNS that is not at least partially linked to microglial (dys)function. Microglia can be described in terms of their morphological appearance and their expression of various microglial antigens, which may be indicative of different functional states. However, ascribing functions merely based on morphology and/or antigen expression may result in oversimplifications, such as M1/M2 (e.g., [1]). This could be improved by including transcriptomic signatures as additional criteria.

Morphologically, the following four microglial phenotypes can be distinguished: resting, activated, amoeboid, and dystrophic microglia (Table 1, Figure 1). A description of microglial morphology is helpful because the cells’ morphology is indicative of their responses to disturbances and inherently connected to their function. However, as we will describe in this review, associating single markers to a distinct phenotype is difficult, since most markers are expressed in several or even all microglial functional states (Figure 1, [2]). Therefore, we give an overview on the available data on microglial markers, with special emphasis on the ionized calcium-binding adapter molecule 1 (IBA1).

## 2. IBA1

Since its first description in 1998 [3], IBA1 has been the most widely used marker for the immunohistochemical analysis of microglia. In contrast to so-called homeostatic microglial markers, such as transmembrane protein (TMEM-)119 or P2RY12, IBA1 is also expressed by peripheral macrophages [4,5,6,7]. Even though IBA1 is often characterized as a typical microglial activation marker [8,9,10], it cannot be used to distinguish between functional microglial phenotypes [2]. IBA1 reliably stains ramified, activated, amoeboid and dystrophic microglia (Figure 1). While the distribution of microglia varies between different brain regions [11], the expression of allograft inflammatory factor 1 (*AIF-1*, gene for IBA1) on microglia does not differ between white matter and grey matter [12].

An increase in IBA1 expression was described in the context of various stimuli. Following the transient occlusion of the middle cerebral artery (MCA) in rats, the IBA1 staining density and intensity were strongly increased in the ischemic core and the ischemic border zone, compared to the non-affected hemisphere [13]. The IBA1 mRNA levels were increased in the peri-infarct tissue and the ischemic core [14], while IBA1-positive cells also expressed the pro-inflammatory cytokines TNFα and interleukin-1α [14].

Increased expression of IBA1 was described multiple times after the application of a high-fat diet. While most studies have focused on the rodent hypothalamus [15,16,17], microglial activation, and therefore an increased appearance of IBA1 in rodent hippocampi was only detected after imitating the effects of a long-term (eight month) Western diet on mice [18]. In contrast, studying the long-term effects of a high-fat, high-carbohydrate diet (HFCD) in the hippocampus of rats, Gzielo et al. [19] showed that the area fraction covered by IBA1-positive cells was reduced after consumption for one year. Furthermore, they were not able to detect any microglial hypertrophy depicting an activated state in HFCD compared to the controls. Since the results differ depending on the used diet, experiment time and investigated brain region, the impact of microglial-induced inflammation in obesity is discussed [20].

Formerly thought to be expressed by all microglial subpopulations [2], our group identified areas in the human hippocampus where a localized loss of IBA1 expression was observed [21], whereas other microglial markers, such as GPX1 or P2RY12, were still present. Therefore, we proposed a novel microglial phenotype characterized by the absence of IBA1 (Figure 2). Interestingly, an increase in these IBA1-free areas was associated with hepatic pathologies. Dennis et al. [22], demonstrated a decrease in IBA1 expression in a subtype of hepatic encephalopathy (HE), which they called non-proliferative HE. In rodents, another metabolic condition was shown to have an impact on microglial IBA1 expression: In the cingulate regions of the anterior corpus callosum and the hippocampus, a decrease in IBA1 expression was observed in long-term hypercorticosteronemia [23].

In patients suffering from schizophrenia, co-localization analysis showed significantly fewer microglia that were positive for IBA1 and TMEM119 in the temporal cortex, and also a decrease in the gene expression of several other microglial markers [24]. Recently, a loss or decrease of IBA1 expression was described in neurodegenerative diseases. In the striatum, Bulk et al. [25] demonstrated an association between an advanced disease state in patients suffering from Huntington’s disease and IBA1-negative microglia. Furthermore, several groups also demonstrated changes in IBA1 expression in association with the Alzheimer’s disease pathology. IBA1-low populations were detected in the human Alzheimer disease cortex [26], while simultaneously showing increased levels of—amongst others—ferritin, CD74 and CD45. Accumulating microglia that were associated with dense-core amyloid plaques were shown to express high levels of HLA-DR, but much less IBA1 [27].

Using transcriptional single-cell sorting, Keren-Shaul et al. [28] demonstrated significant changes in the rodent gene expression of disease-associated microglia (DAM), a concept that was primarily described in a mouse model of Alzheimer’s disease [28]. Compared with homeostatic microglia, DAM were characterized by the downregulation of *AIF1* and homeostatic genes, such as *P2RY12*, *TMEM119*, *Cx3Cr1*, while *CD74* and *CD68* were upregulated (Supplementary Material in [28]).

The triggering receptor expressed on myeloid cells (TREM)-2 serves as a binding partner for apolipoprotein E (APOE, [29]), and especially the ε4 allele has been shown to act as a risk factor for Alzheimer’s disease [30,31]. We have shown that *TREM2* is the highest upregulated mRNA in microdissected periplaque areas [32]. Furthermore, it was demonstrated that TREM2 increases phagocytosis and secretion of anti-inflammatory cytokines [33]. Lue et al. [34] detected an upregulation of IBA1 in association with TREM2 levels in the temporal cortex of Alzheimer disease patients. Consistent with these results, experiments investigating the deficiency of TREM2 detected lower transcription levels of *IBA1* and a decrease in IBA1-positive microglia. This observation was visible at every timepoint tested and even increased with age [35]. However, since TMEM119 was used as the only second marker, it is unknown if the total microglial numbers were reduced or if microglial cells presented with a loss of both TMEM119 and IBA1.

Although the exact function of IBA1 still needs to be elucidated, its involvement in phagocytosis is an accepted hypothesis [7]. However, experiments using interferon regulatory factor 8 (IRF8)-deficient microglia [36], showed significantly reduced levels of IBA1, but no deficits in phagocytosis. IRF8 is one of the intrinsic factors regulating the transition from a ramified to an activated microglial morphology [37,38].

Considering the various diseases that were associated with a decrease or loss of IBA1, it is reasonable to assume that certain microglial dysfunctions that are associated with a loss of IBA1 may not be pathology specific, but have a broader impact on their role in immune defense and synaptic plasticity. Moreover, while being expressed by all microglial clusters in single-cell transcriptomic analysis, the expression of *AIF* was either downregulated or consistent in clusters, depicting a higher number of AD and MS susceptibility genes [39]. In immune histochemical analysis, one needs to distinguish between a discontinuous expression of IBA1, due to shrinkage of the microglial processes, as described as a hallmark characteristic of senescent microglia [40], and an IBA1-negative microglial phenotype. Studies have not discriminated between a reduction in the total microglial numbers and a decreased expression of IBA1, since many use IBA1 as the sole microglial marker.

In order to immunohistochemically investigate IBA1-negative microglia, one needs to use alternative markers. TMEM119 and P2RY12 are especially useful, since they have been shown to be expressed solely by microglial cells and not by infiltrating macrophages [41,42,43]. Both have been described as primarily homeostatic markers, showing a decrease in their expression when microglia become activated. However, more recent studies have suggested a picture of greater complexity.

## 3. TMEM119

TMEM119 has been shown to be stably and specifically expressed in microglia [41,44], and it can be used to distinguish microglia from resident and infiltrating macrophages [41]. In human brains, at least 50% of the IBA1-positive microglial population were positive for TMEM119 [45], while rodent studies even demonstrated an expression of 98% of CD45lowCD11b+ cells in adult animals [41]. Despite being known as a homeostatic marker [45], several studies have shown a stable expression of TMEM119 also in response to injury and inflammatory conditions. In spinal cord injuries, TMEM119-positive cells also stained positive for typical activation markers, such as MHCII and CD68, detecting a pro-inflammatory activation of microglial cells [46]. However, necrotic lesions of cerebral infarctions and demyelinating lesions of multiple sclerosis were devoid of TMEM119 expression [45]. In contrast, MHCII-positive cells in non-active white matter lesions stained positive for not only IBA1 and P2RY12, but also TMEM119 [12]. In a case exhibiting pronounced microglial activation following hypoxic brain damage, the cells did not stain positive for TMEM119 (Figure 3). However, in a human brain sample containing metastatic adenocarcinoma, TMEM119 was positive, even in amoeboid microglia in close vicinity to the neoplastic tissue (Figure 1).

In Alzheimer’s disease brains, the mRNA levels of *TMEM119* were elevated, although no difference was observed in immunohistochemical analyses [28,45]. In contrast, Kenkhuis et al. [47] demonstrated a microglial subset with an increased expression of the iron storage protein ferritin light chain (FTL) and IBA1, while exhibiting a decrease in TMEM119 and P2RY12 expression. This microglial subset presented a morphologically dystrophic phenotype.

Interestingly, PU.1, which was proposed to be a key transcription factor for regulating TMEM119 expression [45], also acts as an upstream regulator of TREM2 [48], suggesting a functional role of TMEM119 in the pathological changes associated with Alzheimer’s disease. Genetic targeting of *TREM2* induced the restored expression of homeostatic microglial markers, such as *TMEM119* and *P2RY12*, concomitant with the G-protein-coupled receptor (*GPR)-34* [49], which is necessary for keeping microglial morphology in a homeostatic non-phagocytic phenotype [50].

## 4. P2RY12

The G-protein-coupled purinergic receptor P2RY12 [51] is specific for parenchymal microglia [52] and shows a persistent expression during the lifespan of microglia [53]. Satoh et al. [45] reported only sporadic expression of P2RY12 in microglia, and that it might not be usable as a generalized microglial marker. In our own experience, decreasing P2RY12 expression was noted in connection with prolonged formalin storage.

Known to be involved in platelet aggregation in the blood, P2RY12 has numerous functions in human microglia. On the one hand, a higher P2RY12 density was detected at microglial membranes directly contacting neuronal somata, suggesting an important role in microglial–neuronal communication [52]. On the other hand, the density of P2RY12 is modified depending on the microglial activation state [54]. Stimulation of P2RY12 triggers the initial extension of microglial processes towards the site of injury [55,56]. However, subsequent downregulation of P2RY12 induces the retraction of microglial processes and therefore leads to morphological activation [57]. Indeed, Sieger et al. [58] demonstrated that the knockdown of P2RY12 in zebrafish led to a complete block of the microglial response to injury. Due to the decrease in its expression in an activated microglial status, P2RY12 is usually described as a homeostatic marker [53,59]. However, Zravy et al. [46], detected a return of low numbers of P2RY12-positive cells in the late stages of the injury process in human spinal cord injury.

Chronically active lesions in multiple sclerosis are characterized by an expression that is similar to non-active white matter lesions (P2RY12+ and MHCII+). The microglia appeared ramified, suggesting a certain habituation effect to the inflammatory conditions of the disease [12]. Furthermore, the microglia in normal-appearing MS tissue were characterized by an unaltered expression of P2RY12 and TMEM119, demonstrating a preservation of microglial homeostatic functions and only localized changes [60].

As shown in studies using single-cell RNA sequencing in Alzheimer’s disease and other neurodegenerative conditions, P2RY12 is one of the proteins marking the transition from homeostatic microglia to disease-associated microglia by the downregulation of gene expression [55,61]. A reduction in P2RY12 immunoreactivity in the microglia was observed prior to massive accumulations of phosphorylated tau protein and neurodegeneration in rTg4510 mouse brains, despite a progressive increase in the total microglial population [62]. Human studies also detected a decrease in P2RY12 in senescent microglia [53,63]. Interestingly, no decrease in P2RY12 was observed around dense-core plaques in APP23 mice [62]. Similar results were shown by Walker and colleagues, who demonstrated that P2RY12 expression differed depending on the types of plaques or tangles they were associated with [64]. Since, in this study, microglia that were positive for P2RY12 also expressed markers of activation, such as CD68, progranulin and to a limited extent, HLA-DR, the mere homeostatic function of P2RY12 needs to be challenged. In primary organotypic brain slice cultures, ethanol caused a unique immune gene signature with a reduction in Tmem119 and a progressive increase in P2RY12 [65], suggesting a distinct microglial phenotype following ethanol treatment.

## 5. CD74

CD74 is also known as the invariant chain and it is necessary for blocking the peptide-binding site of MHCII molecules during their transport from the endoplasmatic reticulum to the cell surface [66,67,68]. However, it was shown that CD74 expression occurred independently from concomitant MHCII expression, and it is also expressed on non-antigen-presenting cells [69]. CD74 was characterized as a cell surface receptor for the macrophage migration inhibitory factor (MIF, [70,71]). The binding of MIF to CD74 leads to an increased recruitment of macrophages and dendritic cells [72]. In cell culture experiments, microglia treated with MIF showed a significant decrease in interferon-γ (IFNγ) expression. Similarly, CD74-silenced microglial cells presented an elevation in IFNγ levels [73]. Furthermore, CD74 was significantly increased in IFNγ-stimulated cultured human microglia [74], suggesting a feedback mechanism and, therefore, an important role of CD74 in IFNγ signaling.

Indeed, Peferoen et al. [74] suggested that CD74 expression represents a pro-inflammatory state. In rodents, CD74 immunoreactivity was not observed in the hippocampus until three days post-ischemia [75]. However, human microglia in all morphological states show a distinct expression of CD74 (Figure 1, [76]), pointing to a potentially important species difference.

Higher levels of CD74 expression in malignant gliomas are associated with a poor prognosis. The activation of CD74 inhibits microglial migration and therefore, invasion into the tumor [73,77]. This makes it a promising target for restoring microglial function. While CD74 has been further described as one of the most up-regulated molecules in human glioblastomas, it was shown that the expression was restricted to glioma-associated macrophages and was absent in tumor cells, with the latter strongly expressing its ligand MIF [78].

In cases of MS, CD74 was expressed in pre-active and remyelinating lesions [74], and interestingly, blocking CD74 in an experimental autoimmune encephalomyelitis (EAE) model ameliorated the symptoms in mice [79]. Furthermore, higher levels of CD74 in monocytes were observed in patients with MS compared to controls [80]. Single-cell analysis demonstrated an increased expression of *CD74* and *HLA-DR* in MS-associated microglial clusters [81], while a great variability in expression patterns displayed a high inter-individual heterogeneity of microglia in the different disease states. As an example of CD74 expression in neurodegenerative disease, CD74 immunocytochemistry in Alzheimer’s disease patients showed expression within microglial processes in and around senile (neuritic and cored) plaques [76]. While also Yoshiyama et al. [82] detected an increase in CD74 in AD microglia. Dystrophic microglia, which appear to precede tau pathology [83], also stain positive for CD74 (Figure 1). Analyzing cell-type-specific expression patterns in the aging human brain, an upregulation of *CD74* in the microglia was detected, concomitant with the upregulation of *TREM2* and *GPR34* [84].

In conclusion, these findings could suggest a certain state of alertness being expressed by CD74. However, in human samples CD74 was not specific for morphologically activated microglia. Therefore, other markers, such as MHCII and CD68, should be considered for immunohistochemically describing the activation states in microglia.

## 6. MHCII

Microglia expressing MHCII can present processed antigens to CD4-positive (CD4+) T-lymphocytes [27], and microglia upregulate HLA-DR in response to IFNγ stimulation [27]. Upregulation was also described in different pathological conditions, such as MS [85] or traumatic brain injury [86]. Moreover, MHCII gene expression is increased in the aged rodent hippocampus, and even more potently by the addition of a high-fat diet (HFD, [87,88]).

Interestingly, several studies have reported genetic risk variants for Parkinson’s disease (PD) in the human leukocyte antigen (HLA) region encoding MHCII molecules [89,90,91,92,93]. Furthermore, MHCII expression precedes and regulates dopaminergic neurodegeneration in the substantia nigra [94,95,96]. Similar to a case with pronounced activation (Figure 3), MHCII-positive cells co-localized with phosphorylated α-synuclein displayed a lower IBA1 immunoreactivity, hypertrophic cell bodies and an amoeboid morphology. This suggests an active involvement of these cells in the clearing of α-synuclein [94].

In studies characterizing microglial phenotypes in MS, microglia tended towards increased MHCII expression, especially in active lesions [12]. While P2RY12 was significantly lower in cortical IBA1-positive microglia compared to controls. Interestingly, the other homeostatic marker—TMEM119—was not differentially expressed [97].

## 7. CD68

As a lysosomal marker, CD68 is mainly expressed in the soma of ramified microglia. Therefore, the typical extrusions cannot not be observed with CD68 staining ([27], Figure 1), making it a non-suitable marker in the examination of microglial morphology. It is indicative of phagocytic activity [2], and is upregulated in cell culture experiments after treatment with pro-inflammatory cytokines, lipopolysaccharide (LPS) and IFNγ [98]. Hence, the upregulation of CD68 is often observed in functionally activated microglia [99]. As a macrophage marker it is not solely expressed in microglia, but also in other macrophages, neutrophils, and monocytes [100]. CD68 has the ability to act as a scavenger receptor, binding oxidized low-density lipoproteins (oxLDL, [101]).

In an analysis of 16,096 individual microglial transcriptomes, the cluster depicting a distinct upregulation of *CD68* was most enriched for DAM genes, such as *CD74*, *HL-DRB1* and *ITM2B* and was associated with pathologic conditions, such as inflammatory demyelination, ischemia and AD [39]. While aging did not lead to the appearance or disappearance of any clusters in scRNAseq analysis of rodent microglia, two clusters were identified as being enriched. These cells upregulated a number of inflammatory signals, such as the cytokine interleukin 1 beta (*IL1β*) and also CD68 [102]. CD68 expression is also associated with lipofuscin, which accumulates in the microglia with aging.

In Parkinson’s disease, an increased number of amoeboid, CD68-positive microglia were found compared to incidental Lewy body disease and control cases [103], suggesting active phagocytosis. Additionally, CD68 exhibited a similar expression pattern to MHCII.

However, Van Olst et al. [97] demonstrated no increased expression of CD68 in multiple sclerosis, therefore emphasizing the importance of using several markers in order to describe the microglial phenotypes as thoroughly as possible.

In Alzheimer’s disease, CD68 stained positive in bloated cytoplasmic processes of dystrophic microglia [40] and in direct vicinity of amyloid-β plaques [104]. Furthermore, several studies proposed the concept of attempted, but deficient microglial phagocytosis in Alzheimer’s disease [105,106]. Therefore, the occurrence of CD68-positive microglia during the disease progression could display the onset of failing microglial function. In fact, we have shown that dystrophic, much more than activated microglia are present in human AD [83,104,107].

## 8. Summary

The heterogeneity of the microglial population regarding not only cell density but also the transcriptional signature has been noted. Recent experiments using deep sequencing revealed differences not only depending on the brain region, but also on age, gender, and pathology [28,39,99,102,108,109]. These findings reflect the numerous and multi-faceted cell functions of microglia. Further consideration of the current concepts of microglial functional states to evaluate pathology-specific microglial phenotypes is needed, since the cells highly differentiate, even within a particular pathology itself. Histopathologically, it is clear that even neighboring microglial cells can exhibit fundamental differences in their shape and protein expression profile (Figure 3L). Understanding local cues that drive the state of individual microglial cells will be a key endeavor of the future.

## Figures and Tables

**Figure 1 cells-10-02236-f001:**
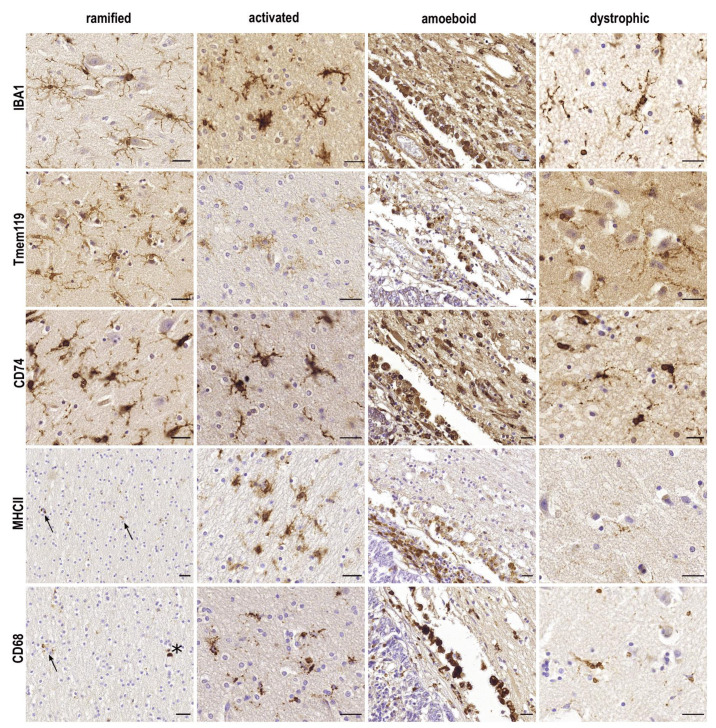
**Comparison of frequently used microglial markers in the typically described morphological phenotypes**. In human tissue from the frontal cortex and hippocampus, most markers show a distinct expression profile in the typically described morphological phenotypes. While IBA1 and CD74 are expressed by all phenotypes, TMEM119 is reduced in activated microglia. Marginal expression of MHCII and CD68 can be observed on parenchymal microglia (↑), while strong expression of perivascular macrophages is shown (*). Amoeboid microglia can be observed on the rim of a metastatic adenocarcinoma in the hippocampus. Scale bars: 25 µm.

**Figure 2 cells-10-02236-f002:**
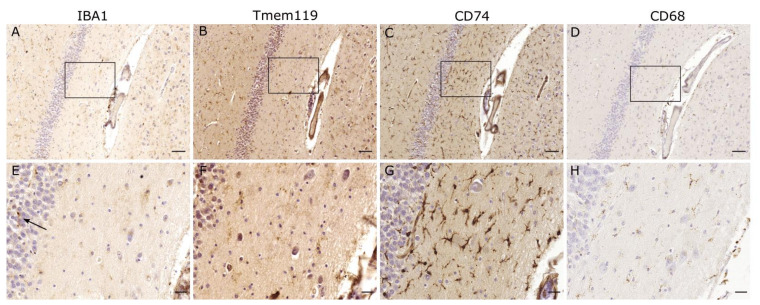
**IBA1-negative microglia.** Regions seemingly devoid of microglia in the IBA1 staining, exhibit positive staining for several other microglial markers such as TMEM119, CD74 and CD68. Occurrence of IBA1-positive microglia at the margin (↑) suggests only localized changes. Scale bars: (**A**–**D**): 100 µm, (**E**–**H**): 25 µm.

**Figure 3 cells-10-02236-f003:**
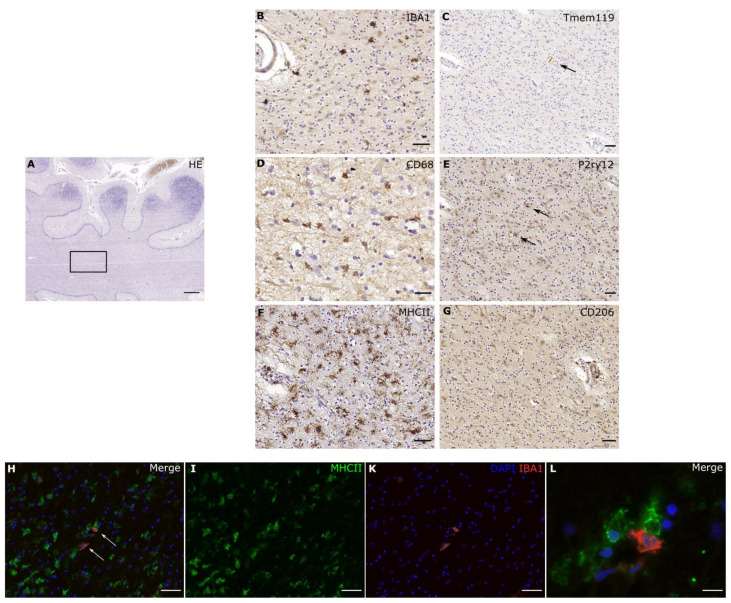
**Pronounced morphological activation of microglial cells.** In a case with hypoxic brain damage after CPR, one can detect pronounced activation and high phagocytic activity. We see a strong expression of MHCII (**F**) and CD68 (**D**), while IBA1 (**B**) is expressed, but the number of IBA1-positive cells is visibly reduced. Homeostatic markers such as TMEM119 (**C**) and P2RY12 (**E**) display only a residual expression (↑). However, CD206 (**G**), a marker for peripheral macrophages, stays negative, therefore a distinct recruitment of non-cns macrophages seems unlikely. In the immunofluorescence, the majority of cells express MHCII, while only two cells express IBA1. Interestingly, microglial cells with different expression profiles are located in direct vicinity from each other. Scale bars: (**A**–**C**), (**E**–**K**): 50 µm, (**D**): 20 µm, (**L**): 10 µm.

**Table 1 cells-10-02236-t001:** Morphological microglial phenotypes.

	Ramified	Activated	Amoeboid	Dystrophic
function	surveillance, synaptic pruning	phagocytosis, antigen presentation	phagocytosis	potential loss of functions
morphology	long ramifications, minute somata	shortened ramification, increased size of cell somata	rounded cell somata, no ramification	swelling and thinning of processes (seemingly fragmented), loss of ramification
markers	IBA1, P2RY12, TMEM119, CD74	IBA1, CD74, CD68, MHCII, ferritin	IBA1, CD74, CD68, MHCII	IBA1, CD74, TMEM119, ferritin
associated diseases	homeostatic conditions	traumatic brain injury, active lesions in MS	ICB, metastases	Alzheimer’s disease

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
