# Peer review of "Beyond Activation: Characterizing Microglial Functional Phenotypes"

_cells, 2021, doi:10.3390/cells10092236_

Round 1
Reviewer 1 Report
- It seems that the authors picked up some microglial marker molecules and correlate them with events affecting their expression pattern. The title of this review is “characterizing functional phenotype” and I would think that, given the recent accumulation of single cell sequence analysis data, consideration of such gene expression profile data is indispensable for describing “functional phenotype” of microglia. And indeed, the abstract mentions about it, but in the manuscript body there is only very limited description on such data. The authors should mention how the microglial cells expressing each of the marker molecules are characterized based on gene expression profile. Otherwise, the title and abstract need to be changed so that they say that this manuscript describes immunohistochemical marker molecules for microglia.
- While it is true that Iba-1 expression is not suitable for distinction of different types of microglia, there are a number of reports showing that Iba-1 expression is enhanced in response to various stimulations, and the authors need to discuss them.
- There are some sentences that I do not understand. (e.g., sentences in line 154-155, line 271-272)
Author Response
Please see the attachement.

Reviewer 2 Report
The review on microglial activation states will be useful and explains the complexity of activation biomarker diversity and expression in the various contexts - in both healthy brain and in disease. The main issues which need attention are writing style and language corrections, as well as clearer explanation in some parts. These are described in detail below:
Abstract line 15, remove the parentheses around ‘single-cell’
Abstract line 20, it’s not clear what ‘life-span’ refers to; is the cells or patient? I assume cells. Please explain this in the text.
Page 1, line 28, first word ‘their’ should be ‘the’
Table 1. column 1, ‘marker’ should be plural ‘markers’
Page 2:
line 53, last word ‘appeared’ should be ‘was seen’ or similar
Line 63, ‘less’ should be ‘fewer’
Line 74-75, regarding APOE, and ‘act as risk factors’ – do you mean that high expression correlates with risk? If so, this should be written.
Fig 1 legend text should include what tissue this is: is it mouse or human brain? Also, label the images or explain what brain regions are shown – the third column tissue images are distinct from the other regions. Last word ‘exhibited’ should ‘shown’.
Page 4: line 127, ‘vicinity’ is better as ‘proximity’
Page 5: Fig 2, delete ‘seemingly’
‘Areas seemingly devoid of microglia in the IBA1 staining’ should be ‘Regions devoid of IBA1-stained microglia…’
Line 143, ‘g-protein-coupled’ should be ’G-protein-coupled’
Line 148: ‘Being known…’ the whole sentence should be rewritten – it is unclear what the message is and whether the functions refer only to PYR12 . I assume the authors are trying to say: ‘Known to be involved in …, P2RY12 has numerous….’
Line 149, ‘…were described.’ Should be ‘…have been described.’
Line 150, delete hence.
Line 161, ‘were characterized…’ change to ‘are characterized…’
Page 6: line 177-178, ‘expressed also markers of…’, change to ‘also expressed markers of…’
Lines 192-195 IFN-gamma is written with the correct Greek gamma symbol the first time, then written as IFN’y’ after that – change the ‘y’ to the Greek symbol.
Line 197, ‘that CD74 expression presents …’, I assume ‘presents should be ‘represents’
Line 204, ‘reinstalling’ should be ‘restoring’
Line 212-213, ‘As an example for a …’, I assume what the authors want to say is ‘As an example of CD74 expression in neurodegenerative disease…’
Page 7, line 235, delete ‘might’
Line 240, should ‘differently’ be ‘differentially’?
Fig 3 title, ‘Pronounced activation’, does this mean ‘Hyperactivation’ or similar? Pronounced activation of what? – all biomarkers? Explain this in the figure legend title
Page 8, line 278, ‘differences depending on brain region, but…’ should be ‘differences not only depending on brain region, but..,’
Round 2
Reviewer 1 Report
The authors addressed the points made by this reviewer.